# Flavonoid-Loaded Biomaterials in Bone Defect Repair

**DOI:** 10.3390/molecules28196888

**Published:** 2023-09-30

**Authors:** Jiali Yang, Lifeng Zhang, Qiteng Ding, Shuai Zhang, Shuwen Sun, Wencong Liu, Jinhui Liu, Xiao Han, Chuanbo Ding

**Affiliations:** 1College of Traditional Chinese Medicine, Jilin Agricultural University, Changchun 130118, China; yjl1104481279@163.com (J.Y.); zlff0429@163.com (L.Z.); ding152778@163.com (Q.D.); zhangshuai4389@163.com (S.Z.); ssw170331@163.com (S.S.); jwlw6803@126.com (W.L.); 2Jilin Agriculture Science and Technology College, Jilin 132101, China; 3School of Food and Pharmaceutical Engineering, Wuzhou University, Wuzhou 543002, China; 4Huashikang (Shenyang) Health Industrial Group Corporation, Shenyang 110031, China; j13364551777@163.com; 5Looking Up Starry Sky Medical Research Center, Siping 136001, China; 18719190727@163.com

**Keywords:** bone defect, biomaterials, flavonoids

## Abstract

Skeletons play an important role in the human body, and can form gaps of varying sizes once damaged. Bone defect healing involves a series of complex physiological processes and requires ideal bone defect implants to accelerate bone defect healing. Traditional grafts are often accompanied by issues such as insufficient donors and disease transmission, while some bone defect implants are made of natural and synthetic polymers, which have characteristics such as good porosity, mechanical properties, high drug loading efficiency, biocompatibility and biodegradability. However, their antibacterial, antioxidant, anti-inflammatory and bone repair promoting abilities are limited. Flavonoids are natural compounds with various biological activities, such as antitumor, anti-inflammatory and analgesic. Their good anti-inflammatory, antibacterial and antioxidant activities make them beneficial for the treatment of bone defects. Several researchers have designed different types of flavonoid-loaded polymer implants for bone defects. These implants have good biocompatibility, and they can effectively promote the expression of angiogenesis factors such as VEGF and CD31, promote angiogenesis, regulate signaling pathways such as Wnt, p38, AKT, Erk and increase the levels of osteogenesis-related factors such as Runx-2, OCN, OPN significantly to accelerate the process of bone defect healing. This article reviews the effectiveness and mechanism of biomaterials loaded with flavonoids in the treatment of bone defects. Flavonoid-loaded biomaterials can effectively promote bone defect repair, but we still need to improve the overall performance of flavonoid-loaded bone repair biomaterials to improve the bioavailability of flavonoids and provide more possibilities for bone defect repair.

## 1. Introduction

Bone is a complex connective tissue with various physiological functions [1,2,3]. Trauma and tumor-related surgery can lead to bone loss and form bone defects [4]. Small bone defects have the possibility of self-repair, but once they exceed the critical defect area, additional intervention is needed to guide and accelerate the healing process [5]. Bone defect healing is a dynamic and complex biological process, and an ideal bone defect implant should conform to the following standards: (1) good biocompatibility; (2) excellent biodegradability; (3) characteristics of bone induction and bone conduction; (4) suitable porosity; and (5) excellent mechanical performance [6,7,8].

Bone transplantation (such as autologous and allogeneic grafts) is the most commonly used and useful method for clinical treatment of bone defects, but it has certain drawbacks, such as insufficient supply of donor tissue and the need for secondary surgery, which increases the risks of infection and surgical costs [9,10]. Therefore, many researchers have designed various biomaterials for bone defect regeneration to overcome these problems, such as metals, ceramics, natural and synthetic polymers, in recent years [11,12,13,14]. Compared to natural polymers, synthetic polymers have poor biocompatibility, which may lead to aseptic inflammatory reactions in bone defect sites [15]. Natural polymers have good biological adhesion, biocompatibility and biodegradability [13], but their mechanical properties are limited and they need to be used together with synthetic polymers to achieve appropriate mechanical strength. Some natural polymers used to prepare bone defect implants include hyaluronic acid, sodium alginate, cellulose, and chitosan. Some synthetic polymers used to prepare bone defect implants include polycaprolactone (PCL) and poly(lactide-co-glycolide) (PLGA) [16,17,18,19].

The excellent properties of bone defect implants designed with the aforementioned polymers can be further enhanced by adding natural bioactive agents. Flavonoids are a class of natural polyphenolic multifunctional plant derivatives [20]. Several studies have reported that flavonoids exhibit a range of biological activities, such as antioxidant, anti-inflammatory, antibacterial, anticancer, antiviral and antiapoptotic abilities [21,22,23,24,25,26]. Many flavonoids can promote bone formation and have antiosteoporosis effects by stimulating osteogenic differentiation of mesenchymal stem cells (MSCs) [27,28,29]. In addition, European nutrition studies have shown that daily intake of flavonoids contributes to good bone health [30]. Due to their biocompatibility, they have attracted widespread attention from many biomedical researchers and have been used to improve bone health. This article reviews the role and mechanism of flavonoid-loaded biomaterials in bone defect repair.

## 2. Bone Defects and Healing Process

Bone is the main supporting tissue and plays different roles in the human body, including protecting organs and the nervous system [31]. When suffering from high-energy trauma, bone cancer, osteoporosis, osteomalacia, osteomyelitis, ischemic necrosis and primary tumor resection caused by bone diseases such as atrophic bone non-union, the loss of bone tissue, bone defects will appear in the human body [32]. The repair rate of bone defects depends on different factors (e.g., age, nutrition, infection, the size of the bone defect) [33]. Generally speaking, small bone defects can self-repair and regenerate [34]. On the contrary, when the defect exceeds the critical level, due to issues such as insufficient blood supply and local infection, bone regeneration does not spontaneously occur easily, which can seriously affect the patient’s quality of life. Therefore, additional clinical treatment is needed to promote bone defect healing [6,35].

Bone regeneration relies on an ideal microenvironment. The microenvironment of bone regeneration is very complex. On the one hand, it is the cross action of various cells, including mesenchymal stem cells, immune cells, endothelial cells, osteoblasts and osteoclast, as well as a variety of bioactive factors, which are involved in osteogenesis, angiogenesis and inflammatory regulation [36]. On the other hand, it spans various stages of bone healing (Figure 1), including hematoma, inflammation, fibrous callus formation, intramembrane ossification, endochondral ossification and bone remodeling accompanied by an orderly cascade of anabolism and catabolism [37].

Specifically, at first, blood clots and immune cell migration in the damaged area migrate to remove necrotic components [38]. Next, endothelial cells and fibroblasts gradually infiltrate to form new capillaries, fibrous matrix and granulation tissue [39]. Subsequently, fibroblasts and mesenchymal stem cells proliferate and differentiate into fibrous tissue, which forms soft callus on granulation tissue [40]. Then, bone forms, including intramembrane ossification (IMO) and endochondral ossification (EO). IMO means that mesenchymal stem cells migrate and proliferate to form coagulation, differentiate into osteoblasts and secrete collagen, and blood vessels grow inward to form cortical bone and cancellous bone [41,42], and EO means that mesenchymal stem cells differentiate into chondrocytes and secrete protein matrix, then blood vessels grow inward and osteoblasts invade and replace chondrocytes to form bone tissue [43]. In the final stage, new bone tissue is continuously absorbed and reshaped, forms an orderly bone structure and restores normal function, characterized by replacing mineralized bone and high levels of osteoblast activity, while cartilage tissue develops [44].

## 3. Classification of Bone Defect Repair Matrices

Bone transplantation is currently the most commonly used method for treating bone defects [45,46]. However, its shortcomings cannot be ignored, for example, the limited amount of donor bone in autologous bone transplantation, and it can easily lead to hematoma, deep infection, inflammation, and uncontrolled absorption rate at the donor site [47,48]. Allogeneic bone grafts have limitations such as immune rejection, disease or virus transmission and require methods such as freeze-drying, radiation and acid washing to avoid immune rejection and eliminate any infections [49,50]. Allogeneic bone transplantation, due to different types of antigens, requires manual handling to avoid possible immune rejection reactions after transplantation [51], and carries the risk of disease, virus transmission and infection [52].

In order to solve the above problems, biomaterials have emerged. According to differences in composition, biomaterials are usually divided into metals, ceramics, natural and synthetic polymers [53]. Compared to natural polymers, synthetic polymers have poor biocompatibility, which may lead to aseptic inflammatory reactions in bone defect sites [15]. Therefore, natural polymers stand out because of their good biocompatibility and biodegradability such as chitosan [54], cellulose [55], hyaluronic acid [56], alginate [56], gelatin [57], etc., which are usually prepared into hydrogels, films and nanomaterials [58,59]. However, their mechanical properties are limited, and they need to be used together with synthetic polymers to achieve appropriate mechanical strength, while their biological activities are still limited.

Bioactive scaffolds are used to transport bioactive molecules such as antibiotics, growth factors and drugs [60,61]. They can continuously supply the required drug concentration at the bone defect site to achieve better treatment effects without obvious secondary adverse reactions. In addition, using biomaterials as drug carriers can protect bioactive molecules from degradation and extend drug circulation and retention time [62,63,64]. Examples of bioactive scaffolds include sponges [65], hydrogels [66], films [67], nanofibers [68], and nanoparticles [69]. These scaffolds are usually designed from natural or synthetic polymers such as cellulose, collagen, sodium alginate, hyaluronic acid and so on [70,71,72,73].

## 4. The Biological Activity of Flavonoids

Flavonoids are the largest component of phenolic compounds and are abundant in fruit, vegetables, flowers, stems, roots, leaves, bark, grains and certain beverages [74,75,76]. They are divided into different subclasses based on the substitution mode of ring C, the oxidation state of heterocycles and the position of ring B, mainly flavones, flavonols, isoflavones, flavanones, flavanes, anthocyanin, chalcones and isoflavanes [20,77,78]. Their basic chemical structures and representative compounds are shown in Table 1. Flavonoids have various biological activities, such as antioxidant, antitumor, antithrombotic, anti-inflammatory, antiallergic, antiviral, antimicrobial and immune regulation [79,80,81,82,83,84,85]. In addition, flavonoids are famous for their roles in bone synthesis and metabolism and have the ability to effectively regulate bone cell function [86]. According to reports [87], flavonoids can stimulate the expression of osteogenic transcription factors and markers through various signaling pathways, such as Wnt and MAPK pathways, to promote the differentiation of MC3T3-E1 osteoblasts and MSCs into osteoblasts.

The expected bone healing effect of flavonoids may be due to their anti-inflammatory, antioxidant and antibacterial activities [88]. First, it is reported that flavonoids are significant inhibitors of inflammatory mediators (COX or LOX), which can inhibit neutrophil degranulation and prevent bone absorption through their anti-inflammatory properties on osteoclast precursor somatic cell cells [89]. Secondly, flavonoids can participate in the activation of enzymes through various signaling pathways and gene expression to eliminate reactive oxygen species (ROS) or free radicals [74]. ROS have negative effects on osteoblasts in a variety of ways, such as osteoblast apoptosis and activation of osteoclast differentiation by upregulating receptor activator of nuclear factor-κB ligand (RANKL) [90,91]. The antioxidant activity of flavonoids is roughly related to their hydrogen supply capacity [92]. In addition, antibacterial performance is another key role of flavonoids, which is crucial for their application in bone defect healing [93].

## 5. Application of Flavonoid-Loaded Biomaterials in Bone Defect Repair

Bone defect healing usually requires a longer repair time. The biomaterial loaded with flavonoids not only has the synergistic effect of biomaterials and flavonoids but also can slowly release flavonoids at the bone defect site to prolong drug efficacy. However, there have been few reports on improving the bioavailability of flavonoids through nanoscale composites. The application of polymer biomaterials loaded with flavonoids in bone defect healing is detailed in Table 2.

### 5.1. Hydrogel

Hydrogel is a hydrophilic three-dimensional polymer that can simulate extracellular matrix (ECM) and has good biocompatibility and biodegradability [126]. However, it also has some drawbacks, such as poor mechanical properties in the swelling state, limited biological activity, etc. Its poor mechanical properties are usually overcome by the combination of synthetic polymers and natural polymers. Chenrui Li et al. prepared a composite material for repairing rat skull defects by combining methacrylic acid chondroitin sulfate and gelatin and incorporated baicalin nanocomposites to enhance the biological activity of the composite system. The experimental results showed that the synthesized composite hydrogel had appropriate mechanical properties. Baicalin nanocomposites can significantly regulate the level of sclerotin and enhance osteogenic and angiogenic activities to play a role in bone repair. These effects are achieved by significantly increasing the expression of OPG, OCN, a-SMA and CD31 and inhibiting the levels of sclerotin and RANKL [94]. Therefore, it is necessary to expand the study of flavonoid-loaded hydrogels to provide a study basis for the development of available medical materials.

### 5.2. Fibrous Membrane

Electrospinning is a simple method for preparing nano- or submicron fiber membranes, and has been widely used in drug delivery [127]. The fiber membrane has a high specific surface area that can promote cell adhesion and continuously and controllably deliver drugs at local points [128]. However, some polymers have encountered some obstacles in electrospinning, such as low mechanical strength, low biocompatibility and low biological activity. Some researchers have reported on polymer fiber membranes loaded with flavonoids. Jung Seung Lee et al. used catechin surface modification of PCL nanofiber membranes to enhance their biological function and prepared a multifunctional matrix for repairing severe skull defects in mice. The research results showed that the deposition of catechin hydrates greatly improved the hydrophilicity and biocompatibility of the matrix. At the same time, the presence of catechin coatings enhanced the antioxidant and calcium binding abilities of the membrane to promote stem cell adhesion, proliferation and osteogenic differentiation, and significantly promoted bone formation in critical size skull defects in vivo [95]. Lihua Yin et al. incorporated icariin as a bone-inducing factor into SF/PLCL nanofiber membranes through electrospinning to enhance the biological activity of the membrane. The study showed that icariin was continuously and controllably released in the nanofiber membrane to effectively increase the expression of ALP and promote bone regeneration in rat skull defects [96]. In addition, Hongbin Zhao et al. prepared a novel core-shell fiber membrane loaded with icariin chitosan microspheres using collagen, polycaprolactone and hydroxyapatite as raw materials for the repair of rabbit tibial defects by electrospinning. The research results showed that the prepared membrane had good mechanical properties and biocompatibility as well as good bone induction and conductivity, and it effectively promoted a large quantity of new bone formation in vivo. These positive effects are achieved by regulating the expression of ALP, COL-1, OC, and OPN [97]. Therefore, these studies indicate that flavonoid-loaded fiber membranes can be successfully used as an effective treatment choice for bone defect repair.

### 5.3. Sponges

Sponge is a three-dimensional structural network that allows cell attachment, migration and proliferation with excellent biocompatibility, porous structure and biodegradability. Its clinical application has good feasibility [129]. However, most biological materials from multiple sources have microenvironments different from bone tissue, which limits their application in bone regeneration. Some researchers have reported on polymer sponges loaded with flavonoids. Mei Li et al. improved bone induction in the submucosa of the small intestine by incorporating icariin into sponge. Studies showed that icariin can be continuously released in sponges for 30 days. Due to the presence of icariin, the sponge significantly promoted the regulation of osteogenic differentiation markers (ALP, BSP, OCN), improved angiogenic factor (CD31) levels, and resulted in a higher rate of new bone formation in a mouse skull defect model [98]. DetingXue et al. found that hesperidin can promote osteogenic differentiation of human mesenchymal stem cells by regulating the ERK1/2 and Smad1/5/8 signaling pathways. Therefore, it was combined with gelatin sponge to accelerate the healing of tibial fractures in rats [99]. Research has shown that hesperidin can significantly increase the levels of osteogenic factors (ALP, OCN, Runx-2, COL-1) and promote bone regeneration in vivo. In addition, Jeong Eun Song et al. incorporated quercetin into collagen/hydroxyapatite sponge to enhance bone metabolism and osteogenic differentiation of the scaffold. The results showed that the prepared sponge had good compressive strength and high porosity and could significantly increase the expression of COL-1, OCN, and Runx-2, promoting the repair of rat skull defects [100]. R.W.K. Wong et al. used collagen sponge loaded with naringin, quercetin, or puerarin to treat full-thickness parietal bone defects in rabbits. These scaffolds can promote angiogenesis and increase ALP activity to achieve early bone reconstruction and bone formation [101,102,103]. It can be seen that flavonoid-loaded sponges have good application prospects in bone defect repair.

### 5.4. Microspheres/Nanoparticles

Due to their excellent specific surface area, microspheres/nanoparticles can improve cell adhesion and proliferation [130] and can be used as carriers for drug delivery systems [131]. However, this method often comes with issues such as low encapsulation efficiency. Zuoying Yuan et al. loaded icariin onto MgO/MgCO_3_ particles and encapsulated them in PLGA microspheres, where Mg^2+^ and icariin were continuously released. Due to the addition of icariin, microspheres significantly regulated the levels of ALP, Col-1, RunX-2, OPN and OCN and promoted the repair of rat skull defects [104]. Xue Yang et al. used PCL/PEG-b-PCL microspheres to reduce the sudden release of naringin and promote the repair of rat skull defects. Studies showed that the prepared microspheres can increase the expression levels of Runx-2 and OCN to promote the formation of new bones in vivo [105]. In addition, Yuning Zhou et al. prepared hydroxyapatite bioceramic microspheres loaded with quercetin, which proved its ability to induce osteogenesis and angiogenesis in vivo in a severe-size femoral defect model of rats. The results showed that the prepared microspheres could continuously and effectively release quercetin to significantly improve ALP activity and the levels of osteogenic genes (Runx-2, COL-1, BSP, BMP-2, OPN, OCN, and OPG), activate ERK, p38 and AKT signaling pathways, upregulate the expression of VEGF, ANG-1, TGF-β and bFGF, and downregulate the expression of an osteoclast gene (RANKL) to promote the repair of rat femoral defect [106]. Yuqiong Wu et al. prepared new micro/nano hybrid HAp particles, constructed a sustained-release system for icariin and verified its role in promoting bone defect repair in a rat femoral defect model. The results suggested that icariin can obviously increase the expression of osteogenic genes (Runx-2, ALP, Col-1 and OCN) and angiogenic genes (VEGF and ANG-1) and regulate the AKT signaling pathway to enhance angiogenesis and bone formation in vivo [107]. It can be seen that the addition of flavonoids can effectively enhance the bone repair activity of the microsphere/nanoparticle system, and the nanoparticle/microsphere system also improves the bioavailability of flavonoids to provide a new idea for the application of bone defect repair (Figure 2).

### 5.5. Bone Cement/Bioglass

Bone cement is an injectable biomaterial with good bone conductivity [132]. Bioglass has excellent specific surface area and its application in bone tissue engineering is rapidly expanding [133]. Bone cement/bioactive glass composed of calcium salts can stimulate new bone formation. However, their biological activities, such as promoting angiogenesis and anti-inflammation, are limited. Some researchers reported on bone cement/bioglass loaded with flavonoids. Jiyuan Zhao et al. promoted the repair of mouse skull defects by loading icariin with calcium phosphate bone cement. This stent can effectively increase the levels of ALP, Runx-2, OC, BSP and promote angiogenesis [108]. Yuqiong Wu et al. constructed icariin-loaded calcium phosphate cement for repairing skull defects in ovariectomized rats. On the one hand, the scaffold improved the level of ALP and promoted osteoblast differentiation; on the other hand, the scaffold upregulated OPG expression and inhibited RANKL expression and the formation of osteoclasts. In addition, the scaffold promoted the expression of angiogenic factors such as VEGF and ANG-1 to promote angiogenesis [109]. In addition, Jianguo Huang et al. prepared a dual drug release system consisting of icariin, vancomycin and injectable calcium phosphate cement for repairing radius defects contaminated by Staphylococcus aureus. The research results showed that the prepared system can release icariin and vancomycin for a long time to endow the system with excellent anti-inflammatory and osteogenic activities and has great potential in the treatment of contaminated bone injury or infectious bone diseases [110]. The icariin-doped bioglass prepared by Xingzhi Jing et al. can significantly increase the expression of osteogenesis-related proteins (COL-1, OPN) and angiogenic factors (CD31, VEGF) and significantly induce new bone formation and new angiogenesis in a rat skull cap bone defect model [111]. These studies indicate that flavonoid-loaded bone cement/bioglass exhibits enormous potential in bone defect repair.

### 5.6. Scaffolds

In addition to the abovementioned types of scaffolds, some researchers have also reported on many composite scaffolds loaded with flavonoids that have good bone conductivity and biocompatibility.

Tao Wu, Yunlong Xie, and Yunjia Song et al. combined icariin into hydroxyapatite composite scaffolds for bone defect repair. These prepared scaffolds have good biocompatibility and can slowly release icariin for a long time. They have good bone conduction and osteoinduction effects on bone defect models and can fill the bone defect site early to stimulate new bone formation. These positive effects may be attributed to the ability of icariin to upregulate the levels of osteogenic markers Runx-2, ALP and OCN and activate the Wnt signaling pathway [112,113,114]. Xiaowei Xie, Tianlin Lliu and Yuxiao Lai et al. combined icariin into various composite scaffolds and conducted therapeutic studies on bone defect models. These prepared scaffolds can promote bone repair through active angiogenesis, which can be attributed to the regulatory effect of icariin on VEGF. At the same time, these scaffolds can significantly increase the expression of osteogenesis-related proteins such as Bsp, Runx-2, ALP, OCN, OPN, etc. to promote the bone healing process and ultimately stimulate bone defect repair [115,116,117].

Kuoyu Chen, Zhenzhao Guo and Yanping Zuo et al. combined naringin with composite materials to evaluate and compare their potential for repairing bone defects in vivo. Their research results all showed that the addition of naringin enhanced the osteogenic ability of composite scaffolds, with the potential mechanism being to reduce the levels of inflammatory factors such as IL-6 and enhance the expression of osteogenesis-related factors BMP-2, OPN, OCN, Runx-2 and ALP to promote the proliferation of osteoblasts and accelerate bone tissue reconstruction and repair in bone defect models [118,119,120].

The epigallocatechin-3 gallate-loaded β-tricalcium phosphate (β-TCP) scaffold prepared by Reena Rodriguez et al. has excellent anti-inflammatory and antioxidant activities and can effectively promote the repair of critical skull defects in rats [121]. Zhihu Zhao et al. prepared a silk fibroin–hydroxyapatite composite scaffold loaded with naringenin, which can significantly increase the expression of Runx-2, COL-1 and OSX by activating PI3K/AKT, VEGF, and HIF-1 signaling pathways to enhance the osteogenesis and angiogenesis and repair distal femoral defects in rabbits [122]. Shuhei Tsuchiya et al. prepared titanium dioxide implants loaded with kaempferol, which can increase the expression of Runx-2, OCN, ON, OPN, COL-1 and ALP to promote the repair of femoral defects in rats [123]. Henri Granel et al. developed a bioactive glass polycaprolactone mixed scaffold loaded with fisterone, which achieved slow release of fisterone and significantly increased the expression of ALP, Runx-2, and COL-1 to promote the repair of critical skull defects in mice [124]. Jeong Eun Song et al. designed a silk fibroin–hydroxyapatite scaffold loaded with quercetin that exhibited good mechanical strength. The addition of quercetin significantly increased the expression of col1, OCN and Runx-2 to promote the repair of rat skull defects. However, this effect occurred only in low-quercetin-content scaffolds rather than high-quercetin-content scaffolds [125]. It can be seen that flavonoids have various biological activities, such as anti-inflammatory, antioxidant and promoting angiogenesis, that are beneficial for bone defect repair. More types of flavonoids are worth developing more widely to promote the application process of polymer materials in bone repair.

The above studies have shown that polymer biomaterials loaded with flavonoids have a good promoting effect on bone defect healing. This effect is mainly achieved by regulating various signaling pathways, such as Wnt, p38, AKT and Erk signaling pathways, which can effectively promote the expression of angiogenic factors such as VEGF and CD31 and significantly increase the levels of osteogenesis-related factors such as Runx-2, OCN and OPN. Therefore, polymer biomaterials loaded with flavonoids can be widely used to promote bone defect healing.

## 6. Conclusions

This article reviews the progress of research on flavonoid polymer biomaterials and their mechanisms for promoting bone defect repair. When using flavonoid-loaded scaffolds, the rate of bone repair can be accelerated, and their role involves multiple impact mechanisms.

Although flavonoid-loaded biomaterials can promote bone defect repair, there are still some issues that need to be addressed. Firstly, most of the studies described are in the preclinical stage and their results are very promising. However, these biomaterials require clinical trials. In addition, the bioavailability and solubility of flavonoids are limiting factors in utilizing their characteristics. They can be increased in solubility through microspheres, nanoparticles, self-emulsifying drug delivery systems, liposome vesicles, solid dispersions, inclusion complexes and micelles to improve bioavailability. In fact, only a small number of nanoparticles and microspheres are used to load flavonoids for bone defect repair. At the same time, there are various types of flavonoid-loaded bone repair biomaterials, but the mutual binding of each biomaterial is still rare. For example, microspheres and nanoparticles are widely used to improve the low solubility of flavonoids, and hydrogels and nanofibers are widely used to continuously transport flavonoids. However, there are few reports on combining them as composite carriers to load flavonoids to promote bone repair. Nanogel has the excellent characteristics of both nanoparticles and hydrogels, and it has been widely studied in the field of bone repair, but no report of nanogel combined with flavonoids for bone defect repair has yet appeared. Finally, there are a wide variety of flavonoids. At present, the development and application of flavonoids loaded in bone repair biomaterials are limited to a few types, including flavones, flavonols, isoflavones, and flavanes. However, anthocyanin, chalcones and isoflavanes have not yet been applied. Therefore, the application of flavonoids in bone repair scaffolds needs to be expanded.

Based on the current study, we should improve the overall performance of flavonoid bone repair biomaterials. A flavonoid bone repair biomaterial that meets clinical requirements needs to be prepared by combining multiple research fields such as molecular biology and pharmacology. At the same time, innovation in biomaterials not only needs to improve their promoting effects on bone repair but also needs to enhance the penetration of flavonoids into biofilms and enhance cell phagocytosis of flavonoids.

## Figures and Tables

**Figure 1 molecules-28-06888-f001:**
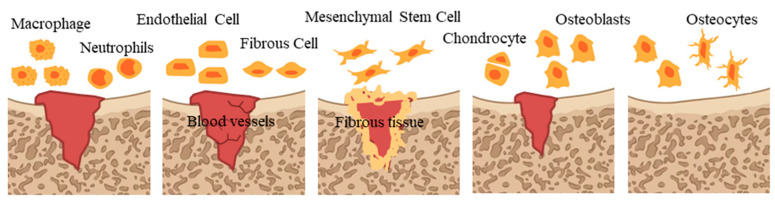
Bone healing process.

**Figure 2 molecules-28-06888-f002:**
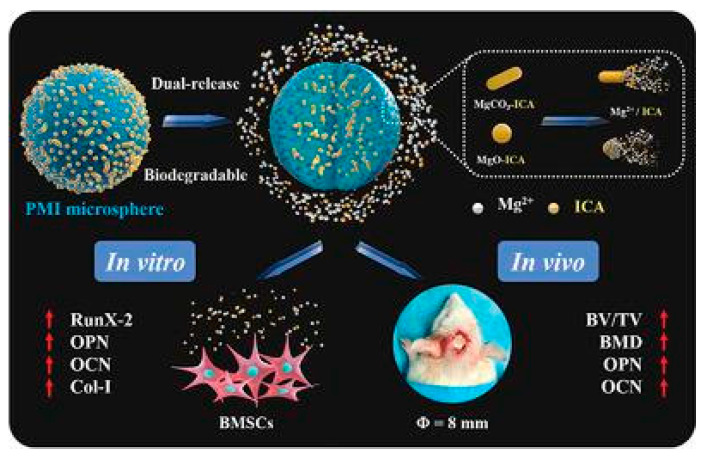
Repair effect of icariin-loaded microsphere on bone defects [105]. (These figures were reprinted with permission.)

**Table 1 molecules-28-06888-t001:** Structure and representative compounds of flavonoid subclasses.

Class	Core Chemical Structure	Typical Compounds
Flavones	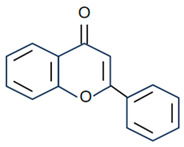	Luteolin, apigenin, hispidulin, chrysin, diosmin, diosmetin, linarin
Flavonols	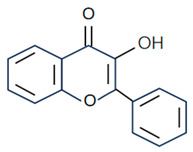	Quercetin, kaempferol, resveratrol, icariin, rutin, linarin, fisetin, myricetin, isoquercitrin
Isoflavones	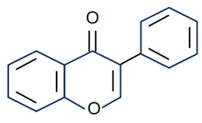	Genistein, daidzein, puerarin
Flavanones	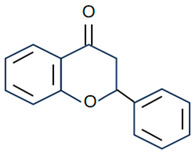	Hesperetin, hesperidin, naringenin, naringin, pinocembrin, dihydroquercetin
Flavanes	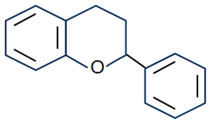	(+)-Catechin, (-)-epicatechin, (-)-epigallocatechin-3-gallate (EGCG)
Anthocyanin	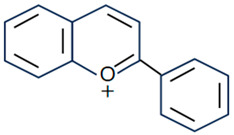	Anthocyanin, delphinidin, cyanidin
Chalcones	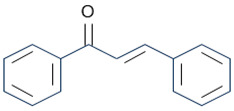	Cardamonin, xanthohumol
Isoflavanes	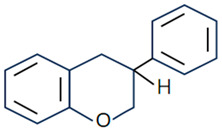	Glabridin

**Table 2 molecules-28-06888-t002:** Application of polymer biomaterials loaded with flavonoids in bone defect healing.

Biomaterials	Biomaterial Composition	Flavonoid Types and Sources	Incorporation/Solubilization Methods	Animal Model	Molecular Mechanism	Reference
Hydrogel	Methacrylated chondroitin sulfate; gelatin	Baicalin (flavones), Scutellaria baicalensis Georgi	Mixing (Solutol HS15 nanocomplex)	Skull defects in rats	Increase the expression of osteoprotegerin (OPG), osteocalcin (OCN), α-smooth muscle actin (α-SMA), and platelet endothelial cell adhesion molecule 1 (CD31); inhibit the levels of sclerosing protein and RANKL	[94]
Fibrous membrane	Polycaprolactone (PCL)	(+)-Catechin (flavanes), Tea leaves, Coffee beans, cocoa	Mixing (hydrate)	Skull defects in mouse	Alleviate oxidative damage	[95]
Fibrous membrane	Silk fibroin (SF); poly(DL-lactide-ε-caprolactone) (PLCL)	Icariin (flavones), Epimedium brevicornum Maxim	Mixing	Skull defects in rats	Increase the expression of alkaline phosphatase (ALP)	[96]
Fibrous membrane	Collagen; polycaprolactone; hydroxyapatite	Icariin (flavones), Epimedium brevicornum Maxim	Mixing (chitosan microspheres)	Tibial defects in rabbits	Increase the expression of ALP, type I collagen (COL-1), osteocalcin (OC) and osteopontin (OPN)	[97]
Sponge	Small intestine submucosa (SIS)	Icariin (flavones), Epimedium pubescens	Absorption	Skull defects in mouse	Upregulate the expression of ALP, bone sialoprotein (BSP) and OCN; increase the level of CD31	[98]
Sponge	Gelatin	Hesperetin (flavanones), citrus fruits	Mixing	Tibial fractures in rabbits	Increase the expression of ALP, OCN, runt-related transcription factor 2 (Runx-2) and COL-1; activate the ERK1/2 and Smad1/5/8 signaling pathways	[99]
Sponge	Collagen, hydroxyapatite	Quercetin (flavonols), synthetic (organic)	Mixing	Skull defects in rats	Increase the expression of COL-l, OCN and Runx-2	[100]
Sponge	Collagen	Naringin (flavanones), grapefruit; quercetin (flavonols), synthetic (organic); puerarin (isoflavones), Pueraria lobata	Mixing	Full-thickness parietal bone defects in rabbits	Promote angiogenesis; increase the activity of ALP	[101,102,103]
Microspheres	Poly(lactide-co-glycolide) (PLGA)	Icariin (flavones), Herba epimedii	Mixing (MgO/MgCO_3_ particles)	Skull defects in rats	Increase the levels of ALP, Col-1, Runx-2, OPN and OCN	[104]
Microspheres	Poly(e-caprolactone) (PCL); poly(ethylene glycol)-block- poly(e-caprolactone) (PEG-b-PCL)	Naringin (flavanones), grapefruit	Mixing	Skull defects in rats	Increase the expression levels of Runx-2 and OCN	[105]
Microspheres	α-Tricalcium phosphate (α-Ca _3_(PO_4_)_2_, α-TCP)	Quercetin (flavonols), synthetic (organic)	Mixing	Femoral defects in rats	Increase the activity of ALP; increase the expression of Runx-2, COL-1, BSP, bone morphogenetic protein 2 (BMP-2), OPN, OCN and OPG; activate the ERK, p38 and AKT signaling pathways; upregulate the expression of vascular endothelial growth factor (VEGF), angiopoietin 1 (ANG-1), transforming growth factor-β (TGF-β) and basic fibroblast growth factor (bFGF); downregulate the expression of RANKL	[106]
Nanoparticles	α-Tricalcium phosphate (α-Ca _3_(PO_4_)_2_, α-TCP)	Icariin (flavones), Epimedium brevicornum Maxim	Absorption	Femoral defects in rats	Promote the expression of Runx-2, ALP, Col-1, OCN, VEGF and ANG-1; regulate the AKT signaling pathway	[107]
Bone cement	Biopex-R	Icariin (flavones), extrasynthese	Mixing	Skull defects in mouse	Increase the levels of ALP, Runx-2, OC and BSP; promote angiogenesis	[108]
Bone cement	Calcium phosphate cement (CPC)	Icariin (flavones), Herba epimedii	Mixing	Skull defects in ovariectomized rats	improve the level of ALP; upregulate OPG expression; inhibit RANKL expression; promote the expression of VEGF and ANG-1	[109]
Bone cement	Calcium phosphate cement (CPC)	Icariin (flavones), Herba epimedii	Mixing	Radius defect contaminated by *S. aureus* in rabbits	Anti-inflammation	[110]
Bioglass	45S5 Bioglass	Icariin (flavones), Herba epimedii	Mixing	Skull defects in rats	Increase the expression of COL-1, OPN, CD31 and VEGF	[111]
Scaffold	Chitosan; hydroxyapatite	Icariin (flavones), Herba epimedii	Mixing	Radial defects in rabbits	Improve the level of ALP	[112]
Scaffold	Hydroxyapatite; alginate	Icariin (flavones), Herba epimedii	Mixing	Radius defects in rabbits	Upregulate the expression of Runx-2, ALP and OCN; activate the Wnt signaling pathway	[113]
Scaffold	Titanium (Ti); glass; hyaluronic acid; chitosan	Icariin (flavones), Herba epimedii	Mixing	Femoral defects in rats	Increase the activity of ALP	[114]
Scaffold	Tricalcium phosphate (TCP)	Icariin (flavones), Herba epimedii	Absorption	Femoral defects in rabbits	Enhance the expression of VEGF	[115]
Scaffold	Siliceous mesostructured cellular foams-poly(3-hydroxybutyrate-co-3-hydroxyhexanoate) (SMC- PHBHHx)	Icariin (flavones), Herba epimedii	Absorption	Skull defects in rats	Increase the expression of Runx-2, ALP and OCN; promote angiogenesis	[116]
Scaffold	Poly(lactic-co-glycolic acid) (PLGA); β-calcium phosphate (β-TCP)	Icariin (flavones), Herba epimedii	Mixing	Distal femoral bone defects in rabbits	Increase the expression levels of BSP, OC, OPN and ALP	[117]
Scaffold	Gelatin; β-tricalcium phosphate	Naringin (flavanones), Citrus fruits	Mixing	Skull defects in rabbits	Enhance the activity of ALP and tartrate-resistant acid phosphatase (TRAP)	[118]
Scaffold	Poly-L-lactide (PLLA)	Naringin (flavanones), Citrus fruits	Mixing (chitosan microspheres)	Periodontal defects in rats	Reduce the expression of interleukin 6 (IL-6)	[119]
Scaffold	Nanohydroxyapatite (nHA); collagen (COL)	Naringin (flavanones), Citrus fruits	Mixing	Skull defects in rats	Increase the expression of BMP-2, OPN, OCN, Runx-2 and ALP	[120]
Scaffold	a-Tricalcium phosphate (a-TCP)	Epigallocate- chin-3-gallate (EGCG) (flavanes), Green tea	Mixing	Skull defects in rats	Anti-inflammation; antioxidation	[121]
Scaffold	Silk fibroin (SF); hydroxyapatite	Naringin (flavanones), Citrus fruits	Mixing	Distal femoral defect in rabbits	Increase the expression of Runx-2, COL-1 and osterix (OSX); activate the PI3K/AKT, VEGF, and hypoxia-inducible factor 1 (HIF-1) signaling pathways	[122]
Scaffold	TiO_2_	Kaempferol (flavonols), vegetables and fruit	Absorption	Femoral defects in rats	Increase the expression of Runx-2, OCN, OPN, COL-1 and ALP	[123]
Scaffold	SiO_2_− CaO bioactive glass−poly(caprolactone) (BG−PCL)	Fisetin (flavones), Vegetables and fruit	Mixing	Skull defects in mice	Increase the expression of ALP, Runx-2 and COL-1	[124]
Scaffold	Silk fibroin; hydroxyapatite	Quercetin (flavonols), synthetic (organic)	Mixing	Skull defects in rats	Increase the expression of Col-1, OCN and Runx-2	[125]

## Data Availability

Data supporting the findings are available from the corresponding authors upon reasonable request.

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
