# Peer review of "Flavonoid-Loaded Biomaterials in Bone Defect Repair"

_molecules, 2023, doi:10.3390/molecules28196888_

Round 1
Reviewer 1 Report (New Reviewer)

Minor editing of English language required
Author Response
Dear reviewer,
Please see the attachment.

Reviewer 2 Report (New Reviewer)
The revision article: “Flavonoid-loaded matrices in bone defect repair” is a relevant addition to the literature, however some issues should be addressed before publishing consideration.
Manuscript is reasonably well organized and wrote, but I suggest considering:
1. Authors should make it clear what kind of revision is considered: narrative or systematic?
2. Describe the selection criteria for articles search.
3. Authors quoted: “Bone defect healing is a dynamic and complex biological process, and an ideal bone defect implant should conform to the following standards: (1) Good biocompatibility; (2) Excellent biodegradability; (3) Characteristics of bone induction and bone conduction; (4) Suitable porosity; (5) Excellent mechanical performance.” but no reference(s) to support it.
4. The phrase: “Bone repair usually involves a longer healing time and requires a certain scaffold to continuously transport flavonoids at the target site. Polymer based materials can continuously provide the required drug concentration at the bone defect site, while protecting bioactive molecules from degradation, prolonging drug circulation and retention time.” should be reworked or relocated. It seems out of context.
5. Authors quoted: “As the main structural organization of the human body, bones play an important role in protecting organs and the nervous system [27]”. Bone is the main supporting tissue (not structural organization) and plays different roles in human body, including protecting....
6. Authors quoted: “The repair rate of bone defects mainly depends on the size of the bone defect.” In fact, the repair rate depends on different factors (e.g.: age, nutrition, infection, among others), including the size of the bone defect.
7. Authors mention “collagen matrix” (L-94, L-101), however, collagen is not the only protein produced. “Fibrous” or “proteic” matrix would be more appropriate.
8. The phrase: “It is believed that a similar process will occur in the case of a large bone defect, except that cell migration requires a longer distance, and larger quality tissue needs to be induced and reconstituted to fill the defect.” should be reworked in addition to the lack of reference.
Minor editing of English language required.
Author Response
Dear reviewer,
Please see the attachment.

Reviewer 3 Report (New Reviewer)
The manuscript"Flavonoid-loaded matrices in bone defect repair" has discussed the polymer biomaterials loaded with Flavonoids". The paper is well written, but some minor changes can improve the quality.
1- the title of the paper is very general, since it uses "matrices". But within the text, the authors have focused on polymeric scaffolds, as stated in the abstract. So, the title can be chosen more specifically.
2- in the abstract, the authors have stated some introductory sentences, whereas in the review papers, different sections of the paper should also be mentioned in the abstract.
3- section 2 (bone defects) and section 3 (the healing process of bone defects) can be merged in one section, not separately.
4- it will be more interesting for the readers if the authors can show the healing process of bone defects, schematically.
5- some paragraphs need to be justified.
The tense of some verbs are future, some past participle and ... Please unify the tenses.
Author Response
Dear reviewer,
Please see the attachment.

This manuscript is a resubmission of an earlier submission. The following is a list of the peer review reports and author responses from that submission.
Round 1
Reviewer 1 Report
The manuscript should be revised to improve several aspects, as described below:
Page 1. Considering that the authors described in the review different matrices, the term “ scaffold” could be replaced by “matrices” or another similar term.
Page 1. Introduction section. Some examples of flavonoids used in bone repair should be added.
Page 2. A section about the activities of the flavonoids, namely anti-inflammatory, anti-oxidant, and so on, and how there are relevant for bone repair should be added.
Page 3. Section 4. Lines 99 to 116 should be summarized. Moreover, lines 117 to 133 should be more useful in contextualizing the review in the introduction. So, they can be moved to the introduction section.
Page 4 , Table 1. It should be useful to have a column related to the composition of the matrices described in table 1. Also, the column serial number is irrelevant and should be removed from the Table. Furthermore, the column called “ Dressing type” is not appropriate. Please consider replaced for “Matrix”. Moreover, the term “other scaffold” can be only “scaffold”.
Page 6 . Considering that the figures from 1 to 5 are illustrative ones, we recommend putting them together in one single Figure with an adequate legend. Moreover, the authors should add figures related to relevant results related to flavonoid-loaded matrices, namely hydrogel, membrane, sponges, microspheres, and so on. Furthermore, it is missing a critical view of the authors along the manuscript facing to the different strategies described in the literature and cited in this review.
Page 7. Section t.3. The term “sponge scaffold” could be misunderstood. Please remove the term” scaffold”. In addition, please avoid beginning the paragraphs with the word “ And”.
Section 6.3. page 9 please add a general discussion about the advantages and disadvantages of the described systems. Moreover, the relevance of the natural polymers combined with flavonoids is cited, but it is not well explored in the manuscript.
Page 10. The extensive description of the examples of the literature should be summarized, and a critical view of the author's should be added.
Page 10- Conclusion section. The future directions and challenges related to the topic do the review is not clear. An appropriate discussion about the different studies on flavonoid-loaded matrices for bone repair and how they can contribute for the advancement of the field should be added.
Page 11. References section. Some references appeared without page and volume. Please check references 43, 60, 73
Some small technical errors were found and it should be corrected.
Author Response
Dear Reviewer1:
Firstly, thank you very much for your comments on our manuscript. These comments are valuable for the revision and improvement of our paper, and also have important guiding significance for our research. We have taken these suggestions into account, revised the manuscript according to the recomendations and responded to related questions. The questions raised by the referees were answered and explained as follows:
Point 1: Page 1. Considering that the authors described in the review different matrices, the term “ scaffold” could be replaced by “matrices” or another similar term.
Response 1: Thank you for your suggestions. We have revised the wording in the manuscript.
Point 2: Page 1. Introduction section. Some examples of flavonoids used in bone repair should be added.
Response 2: Thank you for your suggestions. We have added this section to the introduction.
Point 3: Page 2. A section about the activities of the flavonoids, namely anti-inflammatory, anti-oxidant, and so on, and how there are relevant for bone repair should be added.
Response 3: Thanks for the hard work of the reviewer. The biological activity of flavonoids is described in 5. The Biological Activity of Flavonoids.
Point 4: Page 3. Section 4. Lines 99 to 116 should be summarized. Moreover, lines 117 to 133 should be more useful in contextualizing the review in the introduction. So, they can be moved to the introduction section.
Response 4: Thank you for your suggestions. We have summarized the lines 99 to 116. And we have added content similar to lines 117 to 133 in the introduction.
Point 5: Page 4 , Table 1. It should be useful to have a column related to the composition of the matrices described in table 1. Also, the column serial number is irrelevant and should be removed from the Table. Furthermore, the column called “ Dressing type” is not appropriate. Please consider replaced for “Matrix”. Moreover, the term “other scaffold” can be only “scaffold”.
Response 5: Thanks for the hard work of the reviewer. We have updated Table 2 (formerly Table 1).
Point 6: Page 6 . Considering that the figures from 1 to 5 are illustrative ones, we recommend putting them together in one single Figure with an adequate legend. Moreover, the authors should add figures related to relevant results related to flavonoid-loaded matrices, namely hydrogel, membrane, sponges, microspheres, and so on. Furthermore, it is missing a critical view of the authors along the manuscript facing to the different strategies described in the literature and cited in this review.
Response 6: Thank you for your suggestions. We have updated the figures in the manuscript. And we have redescribed these strategies.
Point 7: Page 7. Section t.3. The term “sponge scaffold” could be misunderstood. Please remove the term” scaffold”. In addition, please avoid beginning the paragraphs with the word “ And”.
Response 7: Thanks for the hard work of the reviewer. We have revised the wording in the manuscript.
Point 8: Section 6.3. page 9 please add a general discussion about the advantages and disadvantages of the described systems. Moreover, the relevance of the natural polymers combined with flavonoids is cited, but it is not well explored in the manuscript.
Response 8: Thank you for your suggestions. We have redescribed these systems and the relevance of the polymers combined with flavonoids.
Point 9: Page 10. The extensive description of the examples of the literature should be summarized, and a critical view of the author's should be added.
Response 9: Thanks for the hard work of the reviewer. We have summarized extensive description of the examples of the literature and added the critical view.
Point 10: Page 10- Conclusion section. The future directions and challenges related to the topic do the review is not clear. An appropriate discussion about the different studies on flavonoid-loaded matrices for bone repair and how they can contribute for the advancement of the field should be added.
Response 10: Thank you for your suggestions. We have rewritten the conclusion.
Point 11: Page 11. References section. Some references appeared without page and volume. Please check references 43, 60, 73
Response 11: Thanks for the hard work of the reviewer. We have checked references.
Finally, Thank you again for the suggestions of the reviewers, which has given us great guidance. If there are still any questions, we will be patient and reply in a timely manner. Thank you for your comments on our manuscript. These comments are valuable for the revision and improvement of our paper, and also have important guiding significance for our research. We have taken these suggestions into account, revised the manuscript according to these suggestions and responded to questions relevant.
Reviewer 2 Report
The authors review the use of flavonoids for treatinf bone defects.
Flavonoids are described simply as “the largest component of phenolic compounds in plants”. Before listing the various biological activities, the flavonoids should be described in terms of their chemical nature, origin, diversity, etc. Despite referring to the chemical structure (line 151, 154), a typical common chemical structure of flavonoids is not shown. neither are described the existing classifications into types, according to the different variations of heterocycles or the existence (or not) of double bond(s). Flavonoids are treated globally throughout the manuscript, and some examples are just mentioned (baicalin, catechin, icariin, naringin, quercetin, puerarin).
Please check that abbreviations are explained when first named, e.g. ALP is described at line 296.
Section 6 is misleading. The authors mention the poor bioavailability and low solubility of flavonoids as limiting factor for taking advantage of their properties. The strategies for dealing with these issues should be discussed in detail. Apparently, the “delivery through scaffolds” ensures a continuous supply. Is this correct for all cases in Table 1? Table 1 does not describe the flavonoid types, nor the matrices materials nor the incorporation/compatibilization methods.
Figures describing the flavonoids delivery process are not entirely accurate. However, Figure 2 is completely wrong. The illustrated fibrous membrane does not match the structure of electrospun scaffolds mentioned in section 6.2, which feature interconnected porosity.
Conclusions are poor and contain confusing phrases that do not capitalize the bibliography review.
Author Response
Dear Reviewer2:
Firstly, thank you very much for your comments on our manuscript. These comments are valuable for the revision and improvement of our paper, and also have important guiding significance for our research. We have taken these suggestions into account, revised the manuscript according to the recomendations and responded to related questions. The questions raised by the referees were answered and explained as follows:
Point 1: Flavonoids are described simply as “the largest component of phenolic compounds in plants”. Before listing the various biological activities, the flavonoids should be described in terms of their chemical nature, origin, diversity, etc. Despite referring to the chemical structure (line 151, 154), a typical common chemical structure of flavonoids is not shown. neither are described the existing classifications into types, according to the different variations of heterocycles or the existence (or not) of double bond(s). Flavonoids are treated globally throughout the manuscript, and some examples are just mentioned (baicalin, catechin, icariin, naringin, quercetin, puerarin).
Response 1: Thank you for your suggestions. We have added this section and created Table 1: Structure and representative compounds of flavonoids subclasses in the manuscript.
Point 2: Please check that abbreviations are explained when first named, e.g. ALP is described at line 296.
Response 2: Thanks for the hard work of the reviewer. We have checked the abbreviations in the entire manuscript.
Point 3: Section 6 is misleading. The authors mention the poor bioavailability and low solubility of flavonoids as limiting factor for taking advantage of their properties. The strategies for dealing with these issues should be discussed in detail. Apparently, the “delivery through scaffolds” ensures a continuous supply. Is this correct for all cases in Table 1? Table 1 does not describe the flavonoid types, nor the matrices materials nor the incorporation/compatibilization methods.
Response 3: Thank you for the reviewer's suggestion. In response to the issues of low bioavailability and solubility of flavonoids, different technologies can be used, such as microparticles, nanoparticle formulations, self emulsifying drug delivery systems, liposome vesicles, solid dispersions, inclusion complexes, and micelles, to deliver flavonoids to the site of action. At the same time, these systems serve as drug repositories and also improve drug permeability and stability. In addition, delivering flavonoids through scaffolds can prolong their release time at the implantation site and improve their bioavailability. Therefore, various carriers can all effectively improve the bioavailability of hydrophobic drugs. According to the reviewer's suggestions, we have supplemented this section in the text and updated Table 2 (formerly Table 1).
Point 4: Figures describing the flavonoids delivery process are not entirely accurate. However, Figure 2 is completely wrong. The illustrated fibrous membrane does not match the structure of electrospun scaffolds mentioned in section 6.2, which feature interconnected porosity.
Response 4: Thanks for the hard work of the reviewer. According to the reviewer's suggestions, we have reorganized the figures.
Point 5: Conclusions are poor and contain confusing phrases that do not capitalize the bibliography review.
Response 5: Thank you for your suggestions. We have rewritten the conclusion.
Finally, Thank you again for the suggestions of the reviewers, which has given us great guidance. If there are still any questions, we will be patient and reply in a timely manner. Thank you for your comments on our manuscript. These comments are valuable for the revision and improvement of our paper, and also have important guiding significance for our research. We have taken these suggestions into account, revised the manuscript according to these suggestions and responded to questions relevant.
Round 2
Reviewer 2 Report
The authors have carried out an extensive bibliographic search work, and many experimental results have been listed, that reflect the positive effects of flavonoids in bone defect repair. However, the concepts and the many works are not well-organized nor presented in a systematic way, as suggested for a review manuscript.
The flavonoid structures and classification were added, as recommended.
The inaccurate schematic figures for flavonoid delivery were replaced by re-printed figures (some of them are not well readable (e.g. Fig 4).
A part of the new Table 2 and Section 6.6 comprises a series of new references called "other scaffolds". It is not clear why they are classified apart: in terms of their structure? matrices? results? flavonoid effect or type? For example, miscrosphere type (ref 103) are described as “Other scaffolds”, but it should have been incorporated in section 6.4 instead.
The strategies for dealing with poor bioavailability and low solubility of flavonoids as limiting factor for taking advantage of their properties are not discussed in detail. Indeed, the meaning of the fourth column of new Table 2 is ambiguous: sometimes the “incorporation method” is referred as the form of the flavonoid, i.e. catechin as a “hydrate” coating for the scaffold surface (ref 81); and sometimes it is referred to a processing technique or a part of a synthesis protocol (e.g. “mixing /immersing” something that remains undefined for the reader).
The Conclusion somehow refer to “stents” features in line 393.